# The Intramolecular Charge Transfer Mechanism by Which Chiral Self-Assembled H_8_-BINOL Vesicles Enantioselectively Recognize Amino Alcohols

**DOI:** 10.3390/ijms25115606

**Published:** 2024-05-21

**Authors:** Rong Wang, Kaiyue Song, Zhaoqin Wei, Yue Sun, Xiaoxia Sun, Yu Hu

**Affiliations:** 1Jiangxi Key Laboratory of Organic Chemistry, Jiangxi Science and Technology Normal University, Nanchang 330013, China; wr19991202@163.com (R.W.); kaiyuesong@163.com (K.S.); weizhaoqin99@126.com (Z.W.); 2State Key Laboratory of Molecular Engineering of Polymers, Shanghai Key Laboratory of Molecular Catalysis and Innovative Materials iChEM, Department of Chemistry, Fudan University, Shanghai 200433, China; sunyue19980307@163.com; 3College of Chemistry, Nanchang University, Nanchang 330031, China

**Keywords:** self-assembled vesicle, amphoteric molecules, H_8_-BINOL, enantioselective fluorescent recognition, ICT

## Abstract

The chiral H_8_-BINOL derivatives R-**1** and R-**2** were efficiently synthesized via a Suzuki coupling reaction, and they can be used as novel dialdehyde fluorescent probes for the enantioselective recognition of R/S-2-amino-1-phenylethanol. In addition, R-**1** is much more effective than R-**2**. Scanning electron microscope images and X-ray analyses show that R-**1** can form supramolecular vesicles through the self-assembly effect of the π-π force and strong hydrogen bonding. As determined via analysis, the fluorescence of the probe was significantly enhanced by mixing a small amount of S-2-amino-1-phenylethanol into R-**1**, with a redshift of 38 nm, whereas no significant fluorescence response was observed in R-2-amino-1-phenylethanol. The enantioselective identification of S-2-amino-1-phenylethanol by the probe R-**1** was further investigated through nuclear magnetic titration and fluorescence kinetic experiments and DFT calculations. The results showed that this mechanism was not only a simple reactive probe but also realized object recognition through an ICT mechanism. As the intramolecular hydrogen bond activated the carbonyl group on the probe R-**1**, the carbonyl carbon atom became positively charged. As a strong nucleophile, the amino group of S-2-amino-1-phenylethanol first transferred the amino electrons to a carbonyl carbocation, resulting in a significantly enhanced fluorescence of the probe R-**1** and a 38 nm redshift. Similarly, S-2-amino-1-phenylethanol alone caused severe damage to the self-assembled vesicle structure of the probe molecule itself due to its spatial structure, which made R-**1** highly enantioselective towards it.

## 1. Introduction

Chirality is a ubiquitous feature in nature; most biological and chemical reactions are accompanied by changes in chirality. In the last decade, more and more fluorescent probes have been designed for enantioselective recognition [1,2,3,4,5,6]. At present, most of the research about the enantioselective recognition of chiral molecules is related to certain chiral amino acids, amino alcohols, α-hydroxycarboxylic acids, and their derivatives. Because of their chiral isomers, they are valuable in biological science and medical research [7,8,9,10,11]. However, the sensitivity, accuracy, environmental factors, and spectral overlap of enantiomers can pose limitations to the application of fluorescence analysis in enantiomeric recognition. The significance of fluorescence enhancement in enantioselective recognition lies in its ability to improve the sensitivity and selectivity of the detection methods [12,13,14,15,16]. By amplifying the fluorescence signal of chiral molecules, enantioselective recognition becomes more efficient and accurate. Overall, the use of fluorescence enhancement in enantioselective recognition greatly enhances the capabilities of analytical chemistry and its applications in various fields [17,18,19,20].

H_8_-BINOL derivatives are well-established enantioselective recognition tools that have been used in a wide variety of applications. In the pharmaceutical industry, the chiral center of the drug molecule can be recognized by the H_8_-BINOL derivative, leading to the preferential formation of one enantiomer over another. This process improves the efficacy and safety of the drug by ensuring that only the desired enantiomer is produced [21,22,23,24]. In environmental analysis, the H_8_-BINOL derivative is used to identify chiral pollutants in water and soil samples. By recognizing the enantiomeric composition of the pollutants, it is possible to assess their environmental impact and develop effective mitigation strategies. In chemical synthesis, the H_8_-BINOL derivative is used to control the stereochemistry of chemical reactions. By guiding the stereochemistry of reactions, it is possible to produce the desired chiral molecules with high enantiomeric purity. Overall, the H_8_-BINOL derivative is a versatile tool. The unique structural features of H_8_-BINOL derivatives also cause them to self-assemble to form vesicles. The presence of aromatic rings in the compound ensures strong π-π interactions, which contribute to the formation of ordered structures [25,26,27]. Additionally, the H_8_-BINOL skeleton also provides ample hydrogen-bonding sites, further facilitating the self-assembly process. Overall, the unique combination of structural features of the H_8_-BINOL derivative allows for spontaneous self-assembly into vesicles, which exhibits important implications for various applications [28,29]. Previously, our group began preliminary studies on the self-assembled structure of H_8_-BINOL and developed novel probes based on the existing studies [30,31,32].

To sum up, many fluorescent probes have been synthesized using the chiral environment of BINOL itself and an aldehyde group to achieve the enantioselective recognition of a wide range of chiral molecules. On the basis of these prior studies, we synthesized R-**1** and S-**1** using H_8_-BINOL, a hydrogenated derivative of BINOL, to provide a chiral environment for the probes. In addition, we introduced a benzene ring with aldehyde and hydroxyl groups at the third position of H_8_-BINOL via a Suzuki coupling reaction. The structural analyses of the crystals showed that it possessed a good π-π force, as well as strong intramolecular and intermolecular hydrogen bonds. The SEM and TEM images showed that it could also perform intermolecular self-assembly to form well-formed vesicles, and the addition of subsequent recognized substances would destroy its vesicle structure. Then, the selective recognition of six chiral molecules by R-**1** was investigated. It was found that R-**1** could enantioselectively recognize 2-amino-1-phenylethanol and enhance the fluorescence of S-2-amino-1-phenylethanol but not R-2-amino-1-phenylethanol, with an enantioselective recognition ratio as high as 10.1 (as shown in Figure 1). (ef = (IS − I0)/(IR − I0), IS, IR: fluorescence intensity of adding 20 eq amino alcohol).

## 2. Results and Discussion

### 2.1. R-***1*** Electron Microscope Images and Crystal Data

In order to achieve the enantioselective recognition of certain chiral amine molecules, two aldehyde-based chiral fluorescent probes based on H_8_-BINOL were designed and synthesized. The synthesis procedure is shown in Figure 1. The introduction of bromine at position 3,3′ of H_8_-BINOL via a simple substitution reaction and an aldehyde group on the benzene ring via a Suzuki coupling reaction not only enhanced the rigidity of the molecule, but also provided a binding site for recognizing chiral molecules. The R-**1** molecule was obtained by removing the methyl group from R-**2** to form a hydroxyl group by BBr_3_. After the selective recognition of six chiral amines using fluorescent probes, it was found that both probes showed a good fluorescence response to S-2-amino-1-phenylethanol but not to the other chiral molecules. And, the intensity of the recognition response of R-**1** (72% yield) was much larger than that of R-**2** (40% yield), so we further explored the enantioselective recognition process of probe R-**1** for R/S-2-amino-1-phenylethanol. The R-**2** fluorescence data can be seen in the Appendix A.

In order to further characterize the specific morphological changes and particle size of the R/S-**1** compounds, we used scanning electronic microscopy (SEM), transmission electron microscopy (TEM), and dynamic laser scattering (DLS).

As shown in Figure 2, R-**1** and S-**1** were prepared as a solution using ethanol as the solvent and sonicated for 30 s. DLS showed that the size of the particles in the solution was about 1.6 μm. In the electron microscopy images, there were many vesicle-type spheres in the solution, which indicated that the composite molecules formed vesicular structures. From the SEM and TEM images in Figure 3, the vesicles of R-**1** are mostly micrometer-scale self-assembled vesicle structures with a large aggregation of multiple vesicles. In contrast, the vesicles of S-**1** are mostly micrometer-scale vesicle structures that lie flat on a plane, with each vesicle relatively intact. We also found that, as more molecules were aggregated, the vesicles formed became larger. Structurally, this molecule had several hydroxyl and aldehyde groups that gave it significant hydrophilic properties. The structure of the cyclohexyl group led to some hydrophobic properties. The interaction between the two ultimately resulted in the formation of amphiphilic molecules that were both hydrophilic and hydrophobic.

In order to further investigate the probe R-**1** molecule and analyze the cause of its supramolecular self-assembly in the SEM images, a single-crystal diffraction test was performed. Fortunately, we were able to grow a single crystal of R-**1**. As shown in Figure 4, the O1 atom of the aldehyde group on the same molecule formed a six-membered cyclic intramolecular hydrogen bond with the H2 atom of the ortho-hydroxyl group on the same benzene ring. At the same time, the probes also formed supramolecular compounds due to the intermolecular hydrogen bonds. These intermolecular hydrogen bonds formed due to the oxygen atoms on two phenol hydroxyl groups of an H_8_-BINOL derivative molecule and the hydrogen atoms on the phenol hydroxyl groups of two H_8_-BINOL biphenyls, respectively. The vertical distance between the substituted phenyl centers of two adjacent molecules was 4.258 A. It was proven that there was a good π-π stacking and hydrogen bonding relationship between the molecules. The dihedral angle between the two axial chiral benzene rings of the H_8_-BINOL molecule was −97.08°, indicating that the chiral plane of the R-**1** molecule was almost vertical. The crystallographic data confirmed that R-**1** formed supramolecular compounds through intramolecular and intermolecular π-π forces and hydrogen bonds, and the main driving factors of its self-assembly into vesicles were numerous intramolecular and intermolecular π-π forces and hydrogen bond effects. The above crystallographic data illustrated that the analysis of this molecule was also of great importance when studying the mechanism of recognition of supramolecular self-assembly and chirality.

### 2.2. Applications of Fluorescence Spectroscopy Testing

As shown in Figure 5, when 2.5 eq of each of the six chiral small molecules in Table 1 was added to the R-**1** probe, the fluorescence intensity of one S-2-amino-1-phenylethanol increased significantly from 1771 to 2469, which was accompanied by a redshift. The fluorescence intensity of several other small molecules did not change much. This indicated that the R-**1** probe could specifically recognize S-2-amino-1-phenylethanol with evident enantioselectivity.

After it was determined that fluorescence recognition had occurred, the difference between the two derivatives was subsequently observed using electron microscopy. The SEM and TEM images shown in Figure 6 reveal that, after the addition of R-2-amino-1-phenylethanol to R-**1**, R-**1** remained vesicular, and the overall morphology did not change significantly. On the other hand, after the addition of S-2-amino-1-phenylethanol, the vesicular morphology of R-**1** basically disappeared, and it developed a structure with an unknown morphology. From these electron microscopy images, it is tentatively clear that R-**1** could enantioselectively recognize S-2-amino-1-phenylethanol, which explains the change in the fluorogram.

In order to test the sensitivity and recognition ability of the probes, titration experiments were then carried out with R-**1** on these two different chiral of amino alcohols, respectively. As it can be seen in Figure 7a, the fluorescence intensity of the probe gradually increased from 1159 to 9056 with the increase in S-2-amino-1-phenylethanol equivalent from 1 eq to 20 eq, which is a nine-fold increase in fluorescence intensity. At the same time, the emission wavelength redshifted from 335 nm to 373 nm by 38 nm. As it can be seen in Figure 7c, the titration results also showed good linearity with R^2^ = 0.99 and a very low detection limit of 1 × 10^−9^ M, indicating that the probe is very sensitive to S-2-amino-1-phenylethanol. Correspondingly, as it can be seen in Figure 7b,c, the intensity of the fluorescence response did not change much with an increasing R-2-amino-1-phenylethanol concentration. This titration experiment confirmed that the probe R-**1** had good enantioselectivity for 2-amino-1-phenylethanol and recognized only S-2-amino-1-phenylethanol. This indicated that the probe can be used as a fluorescence-enhancement probe for S-2-amino-1-phenylethanol with a good linearity and a high sensitivity.

In addition, we performed a titration test to determine the ability of R-**1** to complex with S-2-amino-1-phenylethanol. In Figure 8a,b, the molar fraction of [R-**1**] reaches a minimum when the value of [S]/([S] + [R-**1**]) is 0.5, indicating that the probe R-**1** complexes with S-2-amino-1-phenylethanol at a ratio of 1:1. The complexation constant is 5.1 × 10^5^ M^−1^, which suggests that the probe has a very strong complexation ability. This is important to discuss the ability of R-**1** to specifically recognize S-2-amino-1-phenylethanol.

### 2.3. Exploration of Recognition Mechanisms

It was initially speculated that the recognition mechanism of S-2-amino-1-phenyl ethanol with the fluorescent probe R-**1** was caused by the condensation reaction of R-**1** with S-2-amino-1-phenyl ethanol to form a Schiff base. ^1^H NMR titration experiments were carried out, as it can be seen in Figure 9. After adding 1.0 eq and 2.0 eq S-2-amino-1-phenylethanol to R-**1** for 30 min, we observed a significant change in the ^1^H NMR titration spectra, with a new peak appearing at 8.41 ppm, which was attributed to the generation of H_b_ with C=N. There were also two peaks at 3.86 ppm and 3.70 ppm. These were the H_c_ and H_d_ peaks of methylene, confirming that a Schiff base gradually formed. Meanwhile, the H_a_ peak of the aldehyde group gradually decreased and disappeared at 9.95 ppm. The reaction kinetics were observed using nuclear magnetic resonance titration. After 30 min, only a small amount of the compound was formed, and the conversion rate was 13%. After 15 h, the raw material conversion rate reached 79%, while, after 24 h, the conversion rate reached 85%. The reaction began to slow down.

To verify whether the probe recognition mechanism was responsive to these probes, the Schiff base RS-**4** was synthesized via a condensation reaction between R-**1** and S-2-amino-1-phenylethanol, with a yield of 82% (Figure 2, Appendix A). Then, the fluorescence of the Schiff base RS-**4** was tested under the same conditions, and it was found that RS-**4** had almost no fluorescence (Figure 10). It was confirmed that the enhancement of the fluorescence of R-**1** by S-2-amino-1-phenylethanol was not caused by the formed Schiff base. 

Meanwhile, we also tested the fluorescence time response curve of the probe R-**1** to S-2-amino-1-phenylethanol. As it can be seen from Figure 11, after the addition of S-2-amino-1-phenylethanol, the fluorescence rapidly increased and reached equilibrium in 30 s. After 600 s, the fluorescence did not change significantly. It was proven that the fluorescence change in the reaction was not caused by the production of a Schiff base. If it was a reactive probe, the fluorescence should also have changed significantly with the increase in product concentration over time.

The probe R-**1** consisted of two functionals, an aldehyde group (electron-withdrawing group), and a phenolic hydroxyl group (electron-donating group). The benzene ring containing phenolic hydroxyl and aldehyde groups formed a supramolecular structure through π-π conjugation. Using single-crystal diffraction analysis, the hydrogen atom (electron acceptor) in the hydroxyl group and the oxygen atom (electron donor) in the aldehyde group formed an intramolecular hydrogen bond through a six-membered ring. The intramolecular hydrogen bond activated the carbonyl group, which was conducive to the formation of the enol-like structure, so that the aldehyde group could undergo nucleophilic substitution in the absence of acid catalysis. This also explained the ability of the aldehyde R-**1** and amino alcohol S-2-amino 1-phenylethanol to condense to form a Schiff base without the catalysis of an acid in the nuclear magnetic titration experiments. This is because the phenol hydroxyl group formed an intramolecular hydrogen bond with the oxygen atom in the aldehyde group, which promoted the formation of an enol-like structure and enabled the nucleophilic addition of the aldehyde group under acid catalysis. However, the nucleophilic activity of the amino group on S-2-amino-1-phenylethanol was stronger than that of the alcohol hydroxyl group. Electron transfer occurred first, and the electrons in the amino group quickly transferred to the electron-deficient carbonyl carbocation, resulting in a nucleophilic addition reaction.

In fluorescence recognition, the addition of S-2-amino-1-phenylethanol enhanced the fluorescence while a redshift occurred, which was mainly caused by the intermolecular charge specialized mechanism (ICT). As an electron donor, the lone-pair electrons on the nitrogen atom in the amino group quickly transferred to the electron-deficient carbonyl carbocation, resulting in enhanced fluorescence and a redshift. It can be seen from the fluorescence response time curve that electron transfer occurred within 30 s. Schiff base condensation is a nucleophilic addition reaction between a carbonyl group and a primary amine. Primary amines act as a nucleophilic reagent, and the negatively charged N can react with the positively charged C on the carbonyl group to form the intermediate α-hydroxylamine compound. This is followed by dehydration to form Schiff bases, completing the nucleophilic addition reaction. RS-**4** mainly forms due to the intramolecular hydrogen bonding that first protonates the carbonyl group, and then the amino nitrogen pair of electrons on the amino alcohol acts as an electron donor and rapidly undergoes a charge transfer from the nitrogen atom to the carbonyl carbon site, leading to a nucleophilic addition reaction and the formation of the intermediate α-hydroxylamine compound. Due to the steric hindrance effect, the formation of an intermediate α-hydroxylamine compounds is relatively slow, and they undergo dehydration to obtain the Schiff base RS-**4**. It can be seen from the nuclear magnetic titration data that the Schiff base RS-**4** formed relatively slowly, which is inconsistent with the rapid increase in fluorescence. Therefore, only intermolecular charge transfer occurred in the process of fluorescence recognition, and a Schiff base was not quickly generated (the mechanism is shown in Figure 12) [33,34].

In order to further understand the recognition mechanism of R-**1** with S-2-amino-phenylethanol, we performed density flooding theory calculations. The Gaussian 09W program was used to perform the density flooding of B3LYP/6-31G(d), and the dispersion correction of GD3BJ and factors in the ethanol solvent were taken into account for the calculations. To support the quality of the chosen computational strategy, we also performed calculations on the IR, UV, and fluorescence of R-**1**. In comparison with the experiments, the data were in general agreement, indicating that the calculation method was realistic and reliable (its specific data can be found in the Appendix A). The DFT calculations shown in Figure 13 indicate that the HOMO orbital energy levels are −5.49 eV and −5.48 eV for R-**1** and RS-**4**, respectively, while the LUMO orbital energy levels are −1.81 eV and −1.29 eV, respectively. Moreover, the electron cloud of R-**1** itself is mainly distributed in the LUMO orbital of half of H_8_-BINOL and the HOMO orbital of half of the 3-substituted benzene ring. Once combined with S-2-amino-1-phenylethanol, the electron cloud mainly became concentrated in the LUMO orbitals at the end of half of the carbon/nitrogen double bond and the HOMO orbitals at the end of the H_8_-BINOL, and the bandgap (the difference between the energies of the HOMO and LUMO orbitals) between the ground state energy and the excited state energy increased from 3.68 eV to 4.19 eV, which also indicated that the interaction between the two derivatives led to fluorescence enhancement and a redshift [34,35]. 

Subsequently, we performed structure optimization of the R-**1** monomer as well as the dimer of R-**1** with S-2-amino-1-phenylethanol by the DFT algorithm at the B3LYP-D3(BJ)/6-31g(d) level, taking into account the effect of the ethanol solvent by the SMD solvent model. The π-π overlap of S-2-amino-1-phenylethanol with one phenyl group in the monomer was clearly observed in the dimer. TD-DFT calculations based on the optimized structure were performed at the same computational level and fluorescence emission from the S1 state of the monomer and dimer was obtained. Compared with the monomer, the emission wavelength of the S1 state of the dimer was essentially the same, and the vibronic intensity increased from 0.001 to 0.003. Subsequently, the hole–electron analysis of the emission process of the S1 state was carried out in conjunction with Multiwfn, and the hole–electron isosurfaces were mapped using VMD. As shown in Figure 14, the electrons were mainly distributed in the region where π-π overlap occurred in the monomer, while the holes were mainly distributed in the central fragment of the molecule, and the small Sr indices and large t indices indicated that the fluorescence emission of the dimer was CT charge-transfer excitation (the specific data can be found in the Appendix A).

## 3. Materials and Methods

The drugs used were selected from Aladdin in Shanghai, China; all the solvents were selected from Innochem in Shanghai, China and were used without any pre-treatment. Scanning electron microscopes were taken with a Philips-FEI Tecnai G2S-Twin microscope (FEI, Hillsboro, OR, USA), and head-projection electron microscopes were acquired with an FEI Tecnai F20 (Thermo, Waltham, MA, USA). Crystal data were collected by Smart Apex II, Karlsruhe, Germany. All the NMR data mentioned in this article were measured on a Bruker AM-400WB spectrometer, Karlsruhe, Germany. All the fluorescence data were acquired by a Hitachi F-7100 Fluorescence Spectrophotometer from Hitachi High-Technologies, Tokyo, Japan.

All the NMR and ESI-MS data were placed in auxiliary materials.

### 3.1. Probe Synthesis Steps

#### 3.1.1. Synthesis and Characterization of R/S-**3**

An accurately weighed amount of R/S-H_8_-BINOL (2.0 g, 6.8 mmol) was placed into an eggplant-shaped reaction system. A total of 25 mL of dry CH_2_Cl_2_ was added to dissolve it; then, 10 mL of Br_2_ (1.0 mL, 20.4 mmol) diluted with dry methylene chloride was added dropwise to the system. The reaction was stirred at −30 °C for 4 h and then observed by TLC for completeness. The reaction was quenched with a saturated sodium bisulfite solution (NaHSO_3_) and extracted with CH_2_Cl_2_. The organic phase was washed with saturated saline (NaCl) and dried over anhydrous MgSO_4_. The solvent was rotary evaporated. Separation by column chromatography gave 2.56 g of white solid in an 85.3% yield at EA:PE = 1:35. ^1^H NMR (400 MHz, Chloroform-*d*) *δ* 7.29 (s, 1H), 5.11 (s, 1H), 2.75 (s, 2H), 2.20 (d, *J* = 61.7 Hz, 2H), 1.73 (s, 4H). ^13^C NMR (101 MHz, Chloroform-*d*) *δ* 147.25, 136.79, 132.55, 131.52, 122.25, 107.22, 29.08, 26.92, 22.81, 22.75.

#### 3.1.2. Synthesis and Characterization of R-**2**

To a 100 mL reaction vial was added 20 mL of 1,4-dioxane, dissolved R-**3** (1.0 g, 2.21 mmol), 3-formyl-4-methoxyphenylboronic acid (0.99 g, 5.53 mmol), and Pd(PPh_3_)_4_ (0.51 g, 0.5 mmol), followed by 8 mL of 2 M K_2_CO_3_ solution (dissolved in H_2_O), protected by argon. The reaction was heated at 100 °C for more than 15 h. After the completion of the reaction had been monitored by TLC, the reaction was quenched by the addition of H_2_O, extracted with dichloromethane, washed with saturated brine, and dried with anhydrous MgSO_4_. The solvent was removed by rotary evaporation and separated on an EA:PE = 1:5 column to give 460 mg of light yellow solid in 37% yield. ^1^H NMR (400 MHz, Chloroform-*d*) *δ* 10.48 (s, 1H), 8.07 (s, 1H), 7.86 (s, 1H), 7.16 (s, 1H), 7.03 (s, 1H), 4.94 (s, 1H), 3.96 (s, 3H), 2.79 (s, 2H), 2.30 (d, *J* = 38.1 Hz, 2H), 1.73 (s, 4H). ^13^C NMR (101 MHz, Chloroform-*d*) *δ* 189.85, 161.06, 148.41, 136.96, 131.77, 130.73, 129.33, 124.84, 124.67, 119.83, 111.68, 55.96, 29.82, 29.35, 27.29, 23.14.

#### 3.1.3. Synthesis and Characterization of S-**2**

S-**2** was synthesized by the same method used to synthesize R-**2**, yielding 230 mg of yellow solid in 41% yield. ^1^H NMR (400 MHz, Chloroform-*d*) *δ* 10.48 (s, 1H), 8.07 (s, 1H), 7.86 (s, 1H), 7.16 (s, 1H), 7.03 (s, 1H), 4.94 (s, 1H), 3.96 (s, 3H), 2.79 (s, 2H), 2.30 (d, *J* = 37.9 Hz, 2H), 1.74 (s, 4H). ^13^C NMR (101 MHz, Chloroform-*d*) *δ* 189.69, 160.90, 148.25, 136.80, 131.62, 130.58, 129.18, 124.52, 119.67, 111.54, 55.81, 29.21, 27.15, 22.98.

#### 3.1.4. Synthesis and Characterization of R-**1**

R-**2** (0.5 g, 0.889 mmol) was added to a 100 mL reaction vial at −30 °C, protected by Ar gas, and dissolved by the addition of anhydrous CH_2_Cl_2_. After complete dissolution, BBr_3_ (1.5 mL, 8.89 mmol) was added and reacted for 12 h. After TLC, the reaction was quenched with anhydrous methanol, extracted with CH_2_Cl_2_, washed three times with saturated saline, and dried with anhydrous MgSO_4_. The solvent was removed by rotary evaporation. Separation by column chromatography gave 341 mg of a pale yellow solid in 72% yield at EA:PE = 1:5. ^1^H NMR (400 MHz, DMSO-*d6*) *δ* 10.71 (s, 2H), 10.30 (s, 2H), 7.83 (d, *J* = 2.4 Hz, 2H), 7.70 (dd, *J* = 8.6, 2.4 Hz, 2H), 7.39 (s, 2H), 7.03 (d, *J* = 8.5 Hz, 2H), 6.94 (s, 2H), 2.72 (t, *J* = 6.0 Hz, 4H), 2.23 (dt, *J* = 11.8, 5.8 Hz, 2H), 2.02 (dd, *J* = 15.0, 8.5 Hz, 2H), 1.65 (dt, *J* = 23.2, 7.4 Hz, 8H). ^13^C NMR (101 MHz, Chloroform-*d*) *δ* 196.48, 160.47, 148.03, 137.84, 136.68, 134.08, 131.40, 130.63, 129.64, 124.13, 120.28, 119.47, 117.32, 29.07, 27.03, 22.81, 22.79.

#### 3.1.5. Synthesis and Characterization of S-**1**

S-**1** was synthesized in the same manner as R-**1** to give 363 mg solid in 76% yield. ^1^H NMR (400 MHz, Chloroform-*d*) *δ* 11.02 (d, *J* = 4.4 Hz, 2H), 9.94 (d, *J* = 4.3 Hz, 2H), 7.89–7.73 (m, 4H), 7.17 (d, *J* = 3.9 Hz, 2H), 7.06 (dd, *J* = 8.7, 4.2 Hz, 2H), 4.89 (d, *J* = 4.1 Hz, 2H), 2.82 (q, *J* = 5.5 Hz, 4H), 2.37 (dt, *J* = 10.3, 6.7 Hz, 2H), 2.26 (dd, *J* = 14.1, 8.0 Hz, 2H), 1.87–1.67 (m, 8H). ^13^C NMR (101 MHz, Chloroform-*d*) *δ* 196.47, 160.47, 148.03, 137.84, 136.68, 134.08, 131.40, 130.63, 129.65, 124.14, 120.29, 119.48, 117.32, 29.07, 27.03, 22.81, 22.79.

### 3.2. Fluorescence Experiments

Probe solutions and chiral molecule solutions were prepared at concentrations of 1.0 × 10^−5^ M and 0.05 M. Anhydrous ethanol was chosen as the solvent.

## 4. Conclusions

Two fluorescent probes were designed and prepared via a Suzuki coupling reaction. The benzene ring probe R-**2** was introduced with methoxy and aldehyde groups at the third position of H_8_-BINOL, and then the methyl group was removed to form the hydroxyl-group-bearing probe R-**1**. We found that both the probes could identify S-2-amino-1-phenylethanol with enantioselectivity, but R-**1** could recognize it much better than R-**2** could. The scanning electron microscope images and X-ray analyses showed that R-**1** could form supramolecular vesicles through the self-assembly effect of the π-π force and strong hydrogen bonding. The high enantioselectivity of the R-**1** probe in recognizing S-2-amino-1-phenylethanol was not due to the fact that the two derivatives reacted, thus leading to fluorescence enhancement, as demonstrated by other studies on ^1^H NMR titration and electron microscopy. Instead, the fluorescence reaction mechanism was hypothesized to be an ICT based on the kinetic curves and DFT calculations. The addition of S-2-amino-1-phenylethanol caused the aldehyde group, which had been activated by the hydrogen bonding of the neighboring hydroxyl group, to be subjected to the amino group and undergo only highly efficient and rapid electron transfer, which led to a rapid and pronounced change in fluorescence recognition. This suggests that the enantioselective recognition of these chiral molecules by the probes is potentially relevant in asymmetric synthesis.

## Data Availability

Appendix A to this article can be found online at CCDC 2287198, which contains the supplementary crystallographic data for this paper. These data can be obtained free of charge via www.ccdc.cam.ac.uk/data_request/cif or by emailing data_request@ccdc.cam.ac.uk.

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
