# Peer review of "The Intramolecular Charge Transfer Mechanism by Which Chiral Self-Assembled H8-BINOL Vesicles Enantioselectively Recognize Amino Alcohols"

_ijms, 2024, doi:10.3390/ijms25115606_

Round 1
Reviewer 1 Report
Comments and Suggestions for Authors
In this manuscript, the authors synthesize the chiral H8-BINOL derivatives R-1 and R-2. They use them as novel dialdehyde fluorescent probes for the enantioselective recognition of R/S-2-amino-1-phenylethanol. Additionally, they find both of the two probes could identify S-2-amino-1-phenylethanol with enantioselectivity, and the recognition ability of R-1 was much stronger than that of R-2. Therefore, I would like to recommend the eventual acceptance, after following questions and comments have been addressed satisfactorily.
1. Figure 1, 2, 5, 7, 10 and 11 are too blurry to see clearly and should be improved.
2. In line 111, check whether the size is 16mm.
3. Some English expression needs polishing.
Comments on the Quality of English LanguageShould be improved.
Reviewer 2 Report
Comments and Suggestions for Authors
Dear Editor,
I read the manuscript entitled “The ICT Mechanism of Chiral Self-assembled H8-BINOL Vesicle to Enantioselectively Recognize Amino Alcohols” (Manuscript Nr.: ijms-2982114), submitted by Xiaoxia Sun, Yu Hu and collaborators to the International Journal of Molecular Sciences for possible publication.
In my opinion this study is very interesting. However, the computational part needs a major revision. The Authors forgot to specify the computational details, and thus it is impossible to say anything about the trustworthiness of the used methods. Nonetheless, we can clearly see that the Authors did not perform any conformational equilibria analysis. For example, in Fig. 13, we see the reciprocal orientation of C(=O)H and OH groups does not indicates the formation of an intramolecular H bond, which is very likely to occur. The Authors should provide more connections with the experiments in order to support the quality of the chosen computational strategy (e.g. calculation and comparison of molecular structure of single molecules and complexes, IR, UV-Vis absorption and fluorescence spectra, etc.); the HL energy gap does not tell much (until it is proven that the electronic transitions are H->L or L->H in nature). Calculations could (and thus should) also offer a quantitative picture in terms of the energetics of the reactions and amount of the electron density transferred upon photo excitation/emission. Another major point is that the Authors do not provide any comparison or discussion with the literature. A minor note: it does not make any sense to write the eigenvalues in eV units with 5 decimal digits.
Sincerely.
Reviewer 3 Report
Comments and Suggestions for Authors
The manuscript entitled "The ICT Mechanism of Chiral Self-assembled H8-BINOL Vesicle to Enantioselectively Recognize Amino Alcohols" presents an interesting research. The measurements and experiments seem to be well executed, although the basis of the manuscript is not new (European Journal of Organic Chemistry 2017, 2017(32): 4736, which is cited).
The observed recognition ability of fluorescent probe R-1 is interesting. Since its enantiomer pair, S-1 was also synthesized, it is advisable to perform the experiments with it and show the results. If the heterochiral "complex" is responsible for the enhancement in the fluorescent intensity, the results would serve as strong evidence. It is unclear why these experiments were not performed.
The editing and structure of the manuscript is adequate, however, the English language needs to be carefully reviewed and corrected, it contains too many grammatical and technical language errors, which makes it difficult to understand in some places. Overcomplicated descriptions should be also avoided.
After inserting the results obtained with catalyst S1 and improving the language, the manuscript will be fully suitable for publication.
Comments on the Quality of English LanguageEnglish language needs to be carefully reviewed and corrected, it contains too many grammatical and technical language errors.
Round 2
Reviewer 3 Report
Comments and Suggestions for Authors
The authors improved the quality of the manuscript. After careful revision of the text, it is clear and readily understandable.
I accept the answer regarding the experiments with the other enantiomer of the fluorescent probe.
I recommend reviewing the text once more, because results should be written in past tense, not in present tense. Furthermore, I have highlighted a few sentences that should be reworded and/or reconsidered. The highlighted pdf file is attached.

The quality of the English language has improved greatly, only minor editing is required.
